# 3D Printing of Personalised Carvedilol Tablets Using Selective Laser Sintering

**DOI:** 10.3390/pharmaceutics15092230

**Published:** 2023-08-29

**Authors:** Atabak Ghanizadeh Tabriz, Quentin Gonot-Munck, Arnaud Baudoux, Vivek Garg, Richard Farnish, Orestis L. Katsamenis, Ho-Wah Hui, Nathan Boersen, Sandra Roberts, John Jones, Dennis Douroumis

**Affiliations:** 1Delta Pharmaceutics Ltd., Chatham, Kent ME4 4TB, UK; atabak.ghanizadehtabriz@nottingham.ac.uk; 2CRI Centre for Research Innovation, University of Greenwich, Chatham ME4 4TB, UK; 3Institute of Technology in Measurements and Instrumentation, University of Rouen, 76130 Mont Saint Aignan, France; quentin.gonotmunck@gmail.com (Q.G.-M.); arnaud.baudoux@univ-rouen.fr (A.B.); 4The Wolfson Centre for Bulk Solids Handling Technology, Faculty of Engineering, Science University of Greenwich, Chatham ME4 4TB, UK; v.garg@greenwich.ac.uk (V.G.); r.j.farnish@greenwich.ac.uk (R.F.); 5μ-VIS X-ray Imaging Centre, Faculty of Engineering and Physical Sciences, University of Southampton, Southampton SO17 1BJ, UK; o.katsamenis@soton.ac.uk; 6Drug Product Development, Bristol Myers Squibb, 556 Morris Avenue, Summit, NJ 07901, USA; ho-wah.hui@bms.com (H.-W.H.); nathan.boersen@bms.com (N.B.); sandra.roberts@bms.com (S.R.); 7Bristol Myers Squibb, Reeds Lane, Moreton, Wirral CH46 1QW, UK; john.jones@bms.com

**Keywords:** 3D printing, selective laser sintering, personalised medicines, oral, carvedilol

## Abstract

Selective laser sintering (SLS) has drawn attention for the fabrication of three-dimensional oral dosage forms due to the plurality of drug formulations that can be processed. The aim of this work was to employ SLS with a CO_2_ laser for the manufacturing of carvedilol personalised dosage forms of various strengths. Carvedilol (CVD) and vinylpyrrolidone-vinyl acetate copolymer (Kollidon VA64) blends of various ratios were sintered to produce CVD tablets of 3.125, 6.25, and 12.5 mg. The tuning of the SLS processing laser intensity parameter improved printability and impacted the tablet hardness, friability, CVD dissolution rate, and the total amount of drug released. Physicochemical characterization showed the presence of CVD in the amorphous state. X-ray micro-CT analysis demonstrated that the applied CO_2_ intensity affected the total tablet porosity, which was reduced with increased laser intensity. The study demonstrated that SLS is a suitable technology for the development of personalised medicines that meet the required specifications and patient needs.

## 1. Introduction

Three-dimensional printing has transformed manufacturing capabilities due to its widespread applications in the automotive [1], biomedical, aerospace [2], and pharmaceutical [3] industries, as well as in art [4]. Over the past decade, there has been a growing interest in the use of 3D printing technologies for pharmaceutical applications, and particularly for the design and fabrication of personalised medicines at the point of care. A wide range of 3D printing techniques such as fused deposition modelling (FDM) [5,6,7], material jetting (MJ) [8], selective laser sintering/melting (SLS/SLM) [9], stereolithography (SLA) [10], micro extrusion [11,12], and/or combinations thereof [13] have been employed in pharmaceutical research.

Despite the small scale capabilities, 3D printing presents several advantages over conventional tableting, including the fabrication of dosages with tailored drug amounts, sizes, shapes, and complex geometries, with specific release profiles [14,15]. Hence, 3D-printed dosage forms can be used for designing individualised patient treatments that meet clinical needs, improve patient compliance, and improve the treatment of diseases [16]. The versatility of 3D printing technologies is evidenced by the numerous studies such as the development of paediatric chewable formulations [12,17], bilayer tablets [18], polypills [19,20], or even orally disintegrating tablets [21].

FDM is the most frequently used printing technology for the development of complex tablet designs or compartmental dosage forms with tuneable drug dissolution rates. Nevertheless, FDM largely depends on the extrudability of pharma grade thermoplastic polymers, APIs, and the printability of the formed API loaded filaments, and drug loading can also be a limiting factor [22,23,24]. Regardless of the numerous 3D printing applications in the design of oral solid dosage forms, the use of SLS/SLM technologies for pharmaceutical applications has not been fully exploited. In SLS/SLM technologies, the printing chamber and the powder reservoir of an SLS printer are both heated to a temperature below the melting point (T_m_) or the glass transition (T_g_) of the printable powder. The top layer of powder in the preheated printing chamber is then exposed to a high-power X-Y axis laser beam, which starts to sinter a predetermined 2D pattern in accordance with the product design. The process is repeated, and another thin layer of powder is dispersed once each layer has been successfully completed. The printing is attained by lowering the printing chamber to a predetermined height and rising the reservoir chamber to a predetermined height. The recoating roller/blade can then apply a fresh powder surface on top of the completed layer. A major advantage of SLS is the reduced number of processing steps (e.g., milling) and excipients in comparison to other printing technologies. Those features render SLS simpler, inexpensive, and flexible, with reduced processing times and material losses. Most importantly, its operational simplicity and small footprint renders SLS an appealing printing technology for point-of-care applications [12]. Fina et al. used SLS to investigate the printing of Kollicoat IR and Eudragit L100-55 paracetamol formulations at 5%, 20%, and 35% loadings by weight. The blends were processed at various intensities to avoid drug degradation, and 3% by weight Candurin® gold sheen was added [25]. Depending on the laser intensity, the produced paracetamol tablets can feature various release profiles due to the different tablet porosities that can be obtained for the two polymers. In another study, the printability of paracetamol tablets with polyethylene oxide, Eudragit (L100-55 and RL), and ethyl cellulose was systematically investigated. The experimental findings revealed that the paracetamol release varied based on the polymer grade, laser scanning speed, and the tablet’s designed shape, such as solid cylindrical, gyroid, and bilayer structures [26]. However, the major disadvantage of these studies was the inadequate paracetamol amounts in the printed printlets and the slow dissolution rates that did not comply with pharmacopeia standards.

Recently Gueche et al. [27] investigated the printing of paracetamol with Kollidon VA64/polyamide 12 blends for the formation of fast-disintegrating dosage forms using an SLS printer equipped with a CO_2_ laser printer. The use of different paracetamol grades with large and plate-like particles or thin and needle-like particles affected the porosity and the dissolution rates of the sintered tablets. In addition, the drug–polymer ratio, the drug loading, and the particle size were found to be key critical material parameters.

To our knowledge, there are no reported studies using SLS for the development of personalised dosage forms. Therefore, the aim of this work was to investigate the print-on-demand capabilities of SLS technology for developing personalised dosage forms that meet pharmacopeia specifications in terms of strength, friability, active dose, and dissolution rate. Carvedilol is used to treat heart failure and hypertension, and it was selected as the model drug substance. The printing process parameters were optimised for loadings of 3.125, 6.25, and 12.5 mg respectively.

## 2. Materials and Methods

### 2.1. Materials

Kollidon VA 64 (VA64) was kindly donated by BASF (BASF-Germany) and carvedilol (CVD) was purchased from Tokyo Chemicals (Japan).

### 2.2. Three-dimensional Printing Blend Preparation

Physical blends of VA64 and CVD were prepared at two ratios (96.875/3.125 and 93.75/6.25 weight percent, respectively). The physical blends were mixed using a turbula shaker-mixer (Glen Mills T2F Shaker/Mixer, USA) at 72 rpm for 10 min to ensure blend homogeneity.

### 2.3. Thermal Gravimetric Analysis (TGA)

TGA (TGA Q5000 Thermal instruments, Crawley, UK) was utilised to investigate the thermal stability of the bulk materials. Approximately 2–2.5 mg of polymer and the API samples were carefully weighed and placed into a standard 40 uL aluminium pan. The samples were heated from 25 °C to 400 °C at a heating rate of 10 °C/min. The extracted raw data were analysed via TA Universal Analysis software (Universal Analysis 2000, version 4.5A, TA instruments, UK).

### 2.4. Differential Scanning Calorimetry (DSC)

DSC (Mettler-Toledo 823e, Switzerland) was used to evaluate the thermal behaviour of the bulk materials and the 3D-printed tablets, as well as to investigate the physical state of the CVD within the 3D-printed tablets. Bulk materials and a section of the 3D-printed tablets were carefully weighed. Approximately 2–2.5 mg of material was placed into a 40 µL aluminium pan and crimped promptly. The CVD and the 3D-printed samples were heated from 25 °C to 160 °C at a heating rate of 10 °C/min. The VA64 was heated from −22 °C to 250 °C at a heating rate of 10 °C/min. The extracted DSC thermograms of the bulk materials and the 3D-printed tablets were analysed using STARe Excellence Thermal Analysis software V18 (Mettler Toledo, Greifensee, Switzerland).

### 2.5. Powder Blend Characterisation

#### 2.5.1. Particle Size Distribution

A Mastersizer 2000 (Malvern Panalytical, Malvern, UK) was utilised to investigate the particle size distribution of the blend. The experiment was performed in triplicate.

#### 2.5.2. Flowability Test (Bulk and Tapped Density)

The Carr’s Index, I, of the 3D printing blend was investigated by measuring the tapped and bulk densities of the powder. After the bulk density was measured using a 250 mL graduated cylinder, the powder was subjected to 1250 taps using a tapped density tester (Copley, JVi Series—Model JV 200i, UK) according to USP2 standards. The Carr’s Index of the blend was calculated using the following equation:I = 1 − (V − V_0_) × 100,(1)
where I is the Carr’s index, V_0_ and V are the initial and final (tapped) powder volumes, respectively, of the blend in the measuring cylinder.

### 2.6. Design and 3D Printing of Tablets

Several cylindrical tablet designs were created via SolidWorks software V.31 (Dassault Systems, Waltham, MA, USA) and converted into stl files. The stl files were then transferred into an open-source slicing software, Slic3r 1.2.9, which generated the printing path (G-codes) readable by the printer. The layer height of each slice was set to 0.2 mm. The tablets were printed using a SnowWhite SLS printer (SHAREBOT, Nibionno, Italy) equipped with a 14 W CO_2_ laser. The physical blend was then placed into both feed beds and the print bed. Several horizontal movements of the re-coater ensured the refreshment of the smooth powder surface prior to printing. The blend and the environment temperature were increased to 90 °C and remained stable throughout the printing process. The waiting time, scanning speed, and end environment temperature were set at 15 min, 8000 pps, and 90 °C, respectively, for all experiments. The printing processing parameters are shown in Table 1.

### 2.7. Characterisation of the 3D-Printed Tablets

#### 2.7.1. Hardness Test

The breaking force of six 3D-printed tablets was measured using a Schleuniger 5Y Tablet hardness tester (Pharmatron, Thun, Switzerland).

#### 2.7.2. Friability Test

Twenty 3D-printed CVD tablets were weighed and carefully placed into an EF-2L fibrillatory apparatus (ELECTROLAB, India). The rotational speed was set to 25 rpm for 4 min. After 100 drops, the weights of the tablets were recorded and the friability was calculated using the following equation:F = ((W_i_ − W_f_)/W_i_) × 100,(2)
where F is the friability, W_i_ is the initial weight of the tablets, and W_f_ is the final weight of the tablets after the friability test.

#### 2.7.3. Scanning Electron Microscopy (SEM)

SEM (Hitachi SU8030, Tokyo, Japan) was utilised to investigate the internal microstructures of the 3D-printed tablets, as well as the laser intensity effect on the tablet porosity and permeability. The tablets were kept secured on an aluminium stub with a conductive carbon adhesive tape (Agar Scientific, Stansted, UK). The tablets were then examined via SEM, and images were captured by an electron beam accelerating voltage of 1 KV and a magnification of 30×.

#### 2.7.4. Weight Variation

Ten 3D-printed tablets were carefully weighed using an analytical scale (XSR Analytical Balance, Mettler Toledo, Greifensee, Switzerland). The average weight and percentage of weight variation were then calculated.

### 2.8. X-ray Powder Diffraction (XRD)

XRD was performed to determine the physical states of the bulk materials and the CVD within the tablets after 3D printing. The XRD data were collected using a D8 Advance X-ray diffractometer (Bruker AXS, Karlsruhe, Germany) equipped with a LynxEye silicon strip position-sensitive detector and parallel beam optics. The diffractometer was operated with transmission geometry using Cu Kα radiation at 40 kV and 40 mA, respectively. The instrument was computer-controlled using XRD commander software (Version 2.6.1, Bruker AXS, Germany), and the data were analysed using EVA software (version 5.2.0.3, Bruker AXS, Germany). The samples were placed between foils of 2.5 µm-thick mylar for the measurements. The data were collected between 5–60° 2 θ with a step size of 0.04° and a counting time of 0.2 s per step.

### 2.9. Microfocus Computed Tomography (μCT)

The micro- and macro-porosities of representative 3D-printed tablets at 25%, 40%, and 55% laser intensities was measured. SLS-printed components were also characterised by means of X-ray microfocus computed tomography (μCT). Imaging was performed at the University of Southampton’s μ-VIS X-ray Imaging Centre (www.muvis.org (accessed on 20 August 2023)) using a customised μCT scanner optimised for 3D X-ray histology (www.xrayhistology.org). The system, which is based on Nikon’s XTH225ST system (Nikon Metrology, Castle Donington, UK.), was operated at 110 kVp/73 μA (8 W) without any beam pre-filtration. The source-to-detector and source-to-object distances were 906.7 mm and 30.2 mm, respectively, resulting in a geometric magnification of 30× and a voxel edge size of 10 μm. The 2850 × 2850 dexels detector was binned 2 times, resulting in a virtual detector of 1425 × 1425 dexels. The imaging parameters were as follows: 2001 projections were collected over the 360° rotation, with 4 frames per projection being averaged for each projection to improve the signal to noise ratio.

The reconstructed data were visualised and analysed using Dragonfly software (v. 2022.1.0.1231; Object Research Systems (ORS) Inc., Montreal, QC, Canada, 2020; software available at http://www.theobjects.com/dragonfly). The micro- and macro-pores were segmented using thresholding and manual refinement. The porosity analysis was conducted using the connected components tool applied to the segmented pores volume. The porosity percentage was calculated as the ratio between the “closed and filled” volume of the object and the total volume of the pores, that is, (micro-porosity volume/“closed and filled” volume), (macro-porosity volume/“closed and filled” volume), (total porosity (micro + macro) volume/“closed and filled” volume). “Closed and filled” volume refers to the object volume resulting from the morphologically closed binary mask of the object, followed by filling all internal “voids”. This process captured the space occupied by the object, including the pore space.

### 2.10. In Vitro Dissolution

In vitro dissolution studies were carried out to investigate the release of CVD from the 3D-printed tablets. The release studies were carried out following pharmacopeia guidelines using a Varian 705DS (Varian, USA) dissolution system with an attached paddle apparatus. The release studies were performed at 37 ± 0.4 °C using 900 mL of 0.7% HCl media, adjusted to pH 1.45, using 50% (*w*/*w*) sodium hydroxide. The paddle rotational speed was set to 50 rpm. Three mL of sample media were collected at 5, 10, 20, 30, 45, and 60 min timepoints. The same volume of fresh media was added to each vessel to maintain a constant volume of dissolution media during the release study. The collected samples were then filtered using a 0.2 μm disk. All release studies for the 3D-printed tablets were performed in triplicate.

### 2.11. UV Spectroscopy

All CVD samples were analysed using a UV (Lambda 365, PerkinElmer, Buckinghamshire, UK) meter with a 10 mm matched quartz cell to measure the absorbance of the solution at 247.5 nm [28]. The standard and sample solutions were prepared in distilled water. Calibration curves were made in the range of 1–15 μg/mL (R^2^ = 0.999)

## 3. Results and Discussion

The primary aim of this work was to implement SLS printing technology for the development of personalised dosage forms with the required CVD dose and to meet the pharmacopeia specifications for the recommended CVD release times. Furthermore, the SLS printing process was optimised to ensure that the produced tablets would present adequate strength, friability, and weight variability. SLS could be used as an onsite printing technology for point-of-care services to meet specific patient needs and to help improve clinical outcomes. Prior to 3D-printing optimization, the CVD/VA64 blends were characterised in terms of their thermal properties, particle size distributions, and particle morphologies, which are considered critical material parameters for the printing of SLS tablets.

### 3.1. Thermal Analysis of Plain Polymers

TGA experiments were carried out to evaluate the thermal stability of the bulk CVD and VA64 powders. These powder particles can achieve high temperatures during SLS printing, and the temperatures may vary based on the powder’s features. As shown in Figure 1a, the VA64 presented an initial mass loss of 6% due to moisture content and remained thermally stable up to 290 °C, followed by a rapid mass loss due to polymer degradation. The CVD showed no moisture content and was stable up to 290 °C, followed by a rapid mass loss as a result of degradation caused by the high temperature [29].

As shown in Figure 1b, for the bulk materials, CVD presented a melting endothermic peak at 119.82 °C [30]. The VA64 exhibited a glass transition at 102.72 °C [31]. Previous studies have demonstrated the significant importance of studying the thermal events of bulk materials for print-process optimisation [32]. The environment and build plate temperatures during the printing process must be kept below the melting point or glass transition of the component with the lowest value to avoid melting of the blend prior to laser radiation. Herein, the environment printing temperature was maintained below the glass transition of the VA64 powder.

### 3.2. Powder Characterisation

The particle size distribution of the printing powder is a critical material attribute for the SLS technology as it has been shown to greatly affect flowability and spreading [33]. The use of powder blends with adequate flowability ensures excellent spreading across a surface which, in turn, allows for the formation of uniform layers during printing. As shown in Figure 2, the average particle size distribution for the drug blends was 81.2 μm. As has been reported in previous studies, the particle size of the powder was in a suitable range for printing [34,35]. In addition, the particle morphology of bulk VA64 appeared to be spherical, which is ideal for printing purposes. The flowability of the blends was determined using the Carr’s Index, which was found to be 22.3. The good flowability was related to the flow properties of the VA64. The presence of CVD in the blends had a negligible effect due to its low concentration in the final blends (i.e., 3.125% and 6.25%).

### 3.3. SLS for Printing Personalised Dosage Forms

SLS printing uses laser radiation at temperatures close to the Tm or T_g_ of the polymers to fuse the thermoplastic particles. In a majority of the reported SLS studies where pharmaceutical dosage forms comprising drug and polymer blends were manufactured, the blends did not absorb the laser light and prohibited the sintering process. As a result, Candurin gold sheen varying from 1–3% was added as a binder to improve printability. A major advantage of the current study was its use of a CO_2_ laser (10.6 μm) due to its good absorptivity at the lower applied energies. In addition, CO_2_ printers are cost effective and present better printing precision in comparison to Nd:YAG lasers (λ = 1.064 μm). For the development of the CVD personalised dosage forms, the VA64 was selected as the polymer carrier due to its good flow properties, good sinterability, and high hydrophilicity. The VA64’s flowability ensured adequate powder spreading throughout the layering step while the sinterability was related to the good VA64 absorption from the presence of the -C-O- groups and the powder flow [27,36].

For the print processing parameters, the bed temperature was set at 90 °C, which was below the Tg and Tm of the VA64 and CVD, respectively. The laser speed was selected to be 8000 pps by applying a trial-and-error methodology. It has been previously reported that the laser speed affects the tablet porosity, where a lower speed results in a denser structure, and higher speeds produce less-packed and porous tablets [37]. As shown in Table 1, the laser intensity varied from 25–55% to investigate its effects on the tablet hardness, friability, and dissolution rates of the obtained tablets.

It was observed that by varying the laser intensity and tablet dimensions (e.g., the thickness), the shapes and geometries of the printed structures were not affected. As shown in Figure 3, when the tablets were printed at lower intensities, the external shell presented a lighter density and a light colour, while a dark yellowish colour was observed for the inner cores. The results observed by Cheah et al. [38] showed that the inner cores of CVD/VA64 tablets appeared denser when the intensity was increased from 25 to 55%. Our results were contradictory to those of Cheah as our sintering process, which used a CO_2_ laser, took place through the melting of the VA64 polymer, which captured the neighbouring unmolten particles on the surface of the sintered layers. The formation of non-porous printed tablets with increasing laser intensities was also observed in the SEM analysis, as illustrated in Figure 4. It was observed that the CVD/VA64 particles were completely molten, and the tablet cores appeared entirely fused, with minimal visible voids. By altering the laser intensity and the sintering process, denser tablets can be manufactured, which, in turn, influences the tablet physicochemical and mechanical properties.

To print personalised CVD tablets (Table 2), the active amounts were adjusted by either altering the tablet weight (100–400 mg per tablet) using the same CVD/VA64 batch or by increasing the CVD content from 3.125% to 6.25% in the powder blend. Table 1 presents the average weight, hardness, and friability of tablets printed at various laser intensities. The SLS printing process presented excellent reproducibility and accuracy. The weight variations for the 200 mg and 400 mg tablets were less than 1% and 2%, respectively. The tablet hardness increased from 98–208 N with increasing laser intensities. It was observed that the tablet hardness was also increased with increasing tablet weights. For example, by applying the same laser intensity (40%) and CVD content (12.6 mg), the average hardness for the 400 mg tablets was 105.4 N vs. 35.9 N for the 200 mg tablets.

The tablet friability was inversely proportional to the laser intensity and decreased by increasing the applied radiation. The estimated friability was attributed to the outer tablet walls in which the powder particles were fully sintered. Nevertheless, all tablet batches were easy to handle and did not break freely when transferred.

### 3.4. Physiochemical Characterisation of the 3D-Printed Tablets

DSC and XRD were utilised to investigate the physical state of the CVD within the 3D-printed tablets. As shown in Figure 5 the VA64 XRD showed no peaks due to its amorphous nature while the bulk CVD was in a crystalline state and diffraction peaks appeared at 5.8°, 11.5°, 13.0°, 14.8°, 16.3°, 17.4°, 18.4°, 24.2°, and 29.3°/2θ. The characteristic diffraction peaks of the CVD within the VA64 physical blend (3.125% CVD loading) showed that the drug was in a crystalline form prior to printing. In Figure 5, the X-ray diffractograms of CVD blends post-SLS at different laser intensities show haloes, indicating the absence of CVD crystallinity. To the best of our knowledge, only Madžarević et al. have reported the transformation of an API into an amorphous state as a result of SLS processing for the formation of printed tablets [39]. The findings suggested that for the polymer and drug used in this study, the SLS printing of solid dosage forms may be able to produce an amorphous solid dispersion, increasing the apparent solubility of the API [40]. The applicability of this technology to different drugs and higher drug loads warrants further investigation.

### 3.5. Volumetric Characterisation by Means of X-ray μCT

As shown in Figure 6, μCT analysis was carried out to further investigate the properties of the printed tablets. The experimental findings showed that the total volume, total porosity, and micro- and macro-porosity of the samples varied significantly between the specimens printed at 25% laser intensity and those printed at 40% and 55%, respectively.

Figure 7 shows the total micro- and macro-porosities of the tablets printed at different laser intensities. The total volumes (including the pore spaces) of the samples printed at 55% and 40% laser intensities were 449.25 mm^3^ and 449.76 mm^3^, respectively, with both presenting similar total porosities at approximately 22%. The tablets printed at 25% laser intensity were marginally larger at 466.05 mm^3^ (including the pore spaces), but the total porosity was measured at 28.4%. While the total porosity varied for the different laser intensities, the micro-porosity remained unchanged at approximately 6.6–7.0%. These results indicated that the laser-power-dependent porosity changes were dominated and driven by changes in the macro-porosity, which is better visualised in Figure 8.

A more in-depth characterization of the geometrical, statistical, and physical properties of the printed tablets was not further conducted. However, it is worth noting that an in-depth analysis of the porosity landscape, e.g., for informing computational fluid dynamics (CFD) studies, is possible using various porous media analysis strategies, such as those in [41,42]. In such cases, pores can be handled as “particles”, and specific measures such as individual size, aspect ratio, sphericity, etc., can be extracted. Equally, alternative porous media analysis strategies, such as skeletonization, can be applied to the interconnected porosity fabric (macro-porosity) if properties such as the degree of interconnectivity, tortuosity, and permeability are of interest. Kulinowski et al. introduced such an approach to characterise the topology of pore space [43].

### 3.6. Release Studies

The SLS-printed tablets at different laser intensities and dosage strengths comprising 3.125, 6.25, and 12.5 mg were investigated for their dissolution rates. According to the United States Pharmacopeia, the dissolution rate of CVD active doses should be no less than 80% within 30 min. Figure 9a–e shows the dissolution curves for all the printed tablet batches. For the 3.125 mg strength (100 mg tablet weight) with a 3.125% drug loading (Figure 9), the dissolution rates met the required specifications at all laser intensities. Figure 9b,c shows the dissolution rates for the 6.25 mg (200 mg tablet weight) and 12.5 mg (400 mg tablet weight) strength tablets using the same 3.125% drug loading. As the dose and, therefore, the tablet weight were increased, the dissolution rate and total amount of drug released decreased. For the 6.25 mg tablets, the CVD dissolution rates varied from 40–60% depending on the laser intensity (Figure 9b). For the 12.5 mg tablets, the dissolution rates were approximately 20% after 60 min (Figure 9c).

It was evident that by proportionally increasing the tablet weight (i.e., printing thicker tablets) and the CVD active dose, a slower release was obtained due to the denser structure of the tablet. As a result, the slow VA64 erosion rates determined the CVD dissolution rates. To improve the dissolution rates for the higher dosage strengths, the drug loading in the CVD/VA 64 formulation was increased to 6.25% using a new blend (Table 1). With a total tablet weight of 100 mg, tablets were printed at various laser intensities. As shown in Figure 9d, the tablets printed at 40–55% laser intensities presented slower rates in the first 20 min compared to those printed at 25%. However, the CVD concentrations were greater than 80% after 30 min for all the printed tablets.

Based on these results, the tablet diameter of the 12.5 mg tablet was increased from 10 to 13 mm to maintain the same tablet thickness as the 6.25 mg tablet. It was thought that by maintaining the same surface-to-volume ratio, the dissolution rate of the 12.5 mg tablet would increase [17,44]. As shown in Figure 9e, all three formulations (F12–14) had greater than 80% released in 30 min, meeting the USP specifications.

Table 3 summarises the SLS-printed tablet dissolution, friability, and hardness rates for the various CVD doses processed at different laser intensities. The results demonstrated that SLS can be used for the design and printing of personalised dosage forms. Moreover, it is feasible to produce all CVD dose strengths in one print cycle by varying the laser intensity and the formulation composition.

Overall, the study demonstrated the capabilities of SLS as a key 3D printing technology for the production of solid dosage forms. From the results, it is evident that it can be used for the design and realisation of personalised dosage form by printing tablets at various drug strengths using the same formulation and in one batch. This is not possible for other technologies, such as FDM, where tablet printing is limited by the number of printing nozzles and only tablets of the same strength can be printed each time. Another major advantage of SLS is that it is not limited by the printability of the used excipients and a great number of polymers or lipids can be introduced for the design of the desired dosage form. Moreover, SLS is a simple process due to the limited number of steps for blending the active formulation and printing the designed tablets. Other technologies such as FDM require the production of filaments and the coupling of the extrusion processing for 3D printing. The simplicity of SLS renders it an ideal printing technology for the manufacture of novel oral dosage forms.

## 4. Conclusions

Selective laser sintering has been employed for the design and manufacture of CVD personalised dosage forms at various strengths. It was feasible to print tablets of three different strengths that fulfilled the required pharmacopeia specifications. Modulating the applied laser intensities impacted the tablet quality characteristics, including the tablet hardness, friability, dissolution rate, and total amount of drug released. An increased laser intensity produced a denser tablet, resulting in a slower release rate (i.e., the total amount of drug released over the course of 90 min), increased tablet hardness, and decreased friability.

This SLS study highlights many potential applications and exemplifies the use of this technology for the manufacture of personalised dosage forms, especially for point-of-care services. First, it is feasible that the laser intensity could be used to tailor and modify the release rate of a drug, producing an extended or sustained released effect. Second, SLS was able to manufacture a tablet that did not contain a detectable amount of crystalline drug substance. While the stability of this apparently amorphous drug product was not studied, this technique could be a viable technology to quickly generate amorphous drug products without the need to generate extensive stability or go through time-consuming and cumbersome manufacturing steps. This could hasten the time to clinic for drug substance candidates which are known to require an amorphous form to improve exposure.

## Figures and Tables

**Figure 1 pharmaceutics-15-02230-f001:**
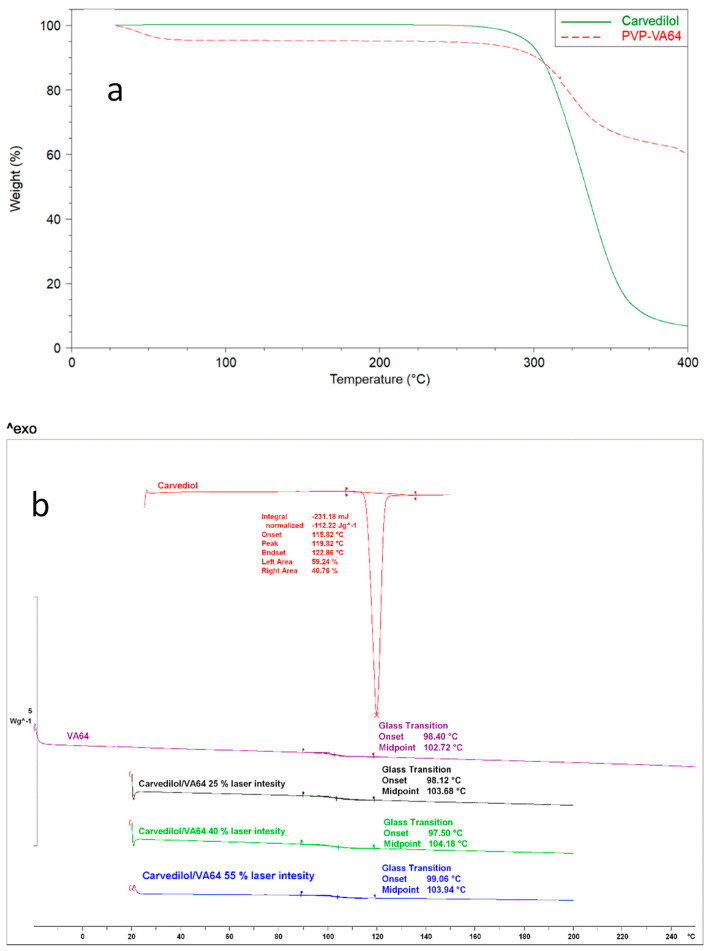
(**a**) Thermogravimetric analysis of the bulk VA64 and CVD powers to investigate their thermal stability, and (**b**) DSC thermograms of the bulk CVD and VA64 powders and the 3D-printed tablets at different laser intensities that varied from 25–55%.

**Figure 2 pharmaceutics-15-02230-f002:**
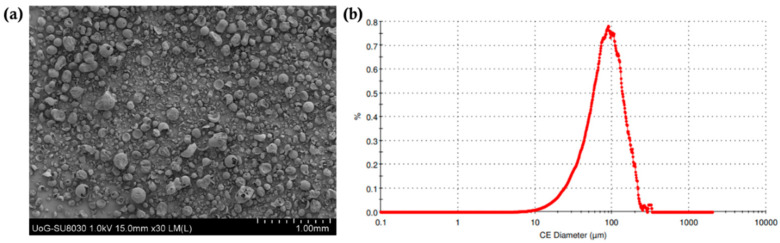
(**a**) SEM image of the CVD/VA 64 physical blend that was used for the printing of formulations 1–8 (30× magnification), and (**b**) the particle size distribution of the blend.

**Figure 3 pharmaceutics-15-02230-f003:**
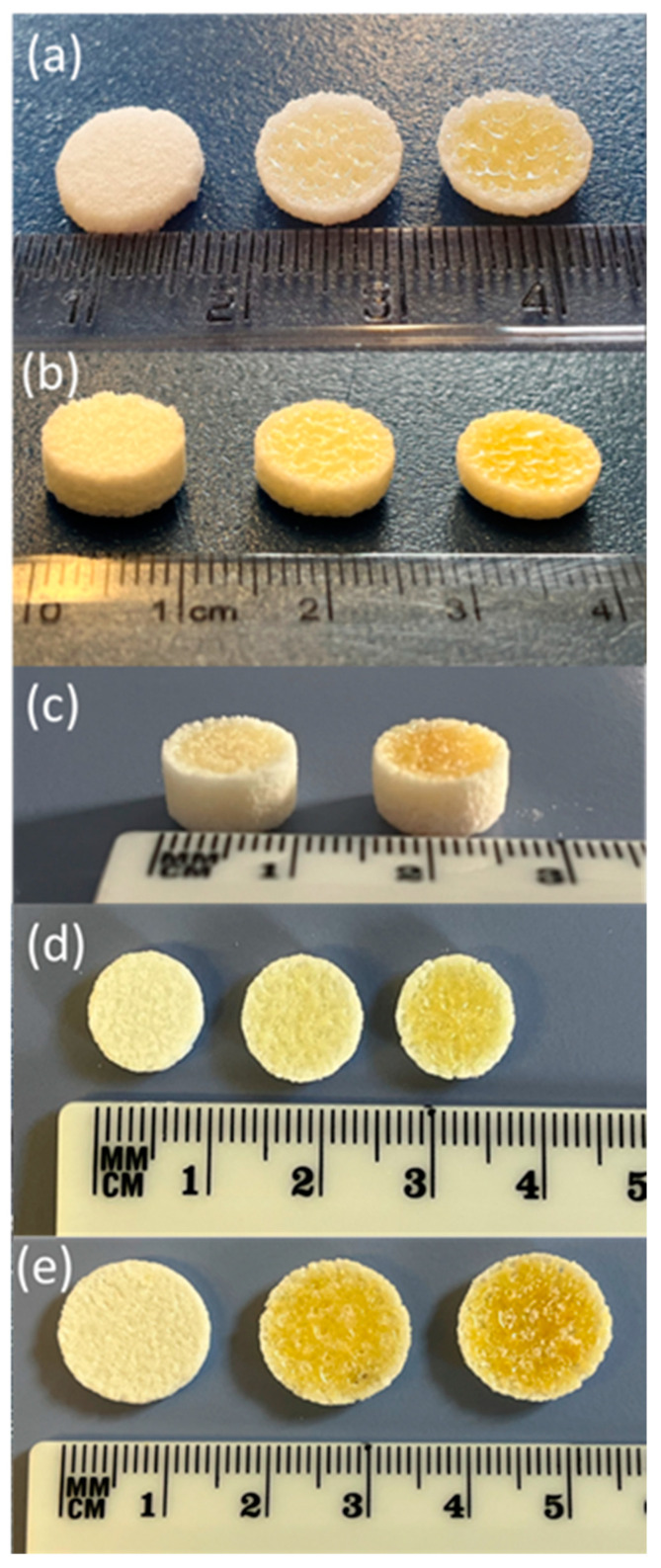
The 3D-printed tablets with different laser intensities: (**a**) 100 mg (laser intensities, from left to right, respectively, of 25, 40, and 55%), (**b**) 200 mg (laser intensities, from left to right, respectively, of 25, 40, and 55%), and (**c**) 400 mg using a 3.125% CVD blend (laser intensities, from left to right, respectively, of 40 and 55%). The 3D-printed tablets printed with different laser intensities: (**d**) 100 mg (laser intensities, from left to right, respectively, of 25, 40, and 55%), and (**e**) 200 mg (laser intensities, from left to right, respectively, of 25, 40, and 55%) with 6.25% loading.

**Figure 4 pharmaceutics-15-02230-f004:**
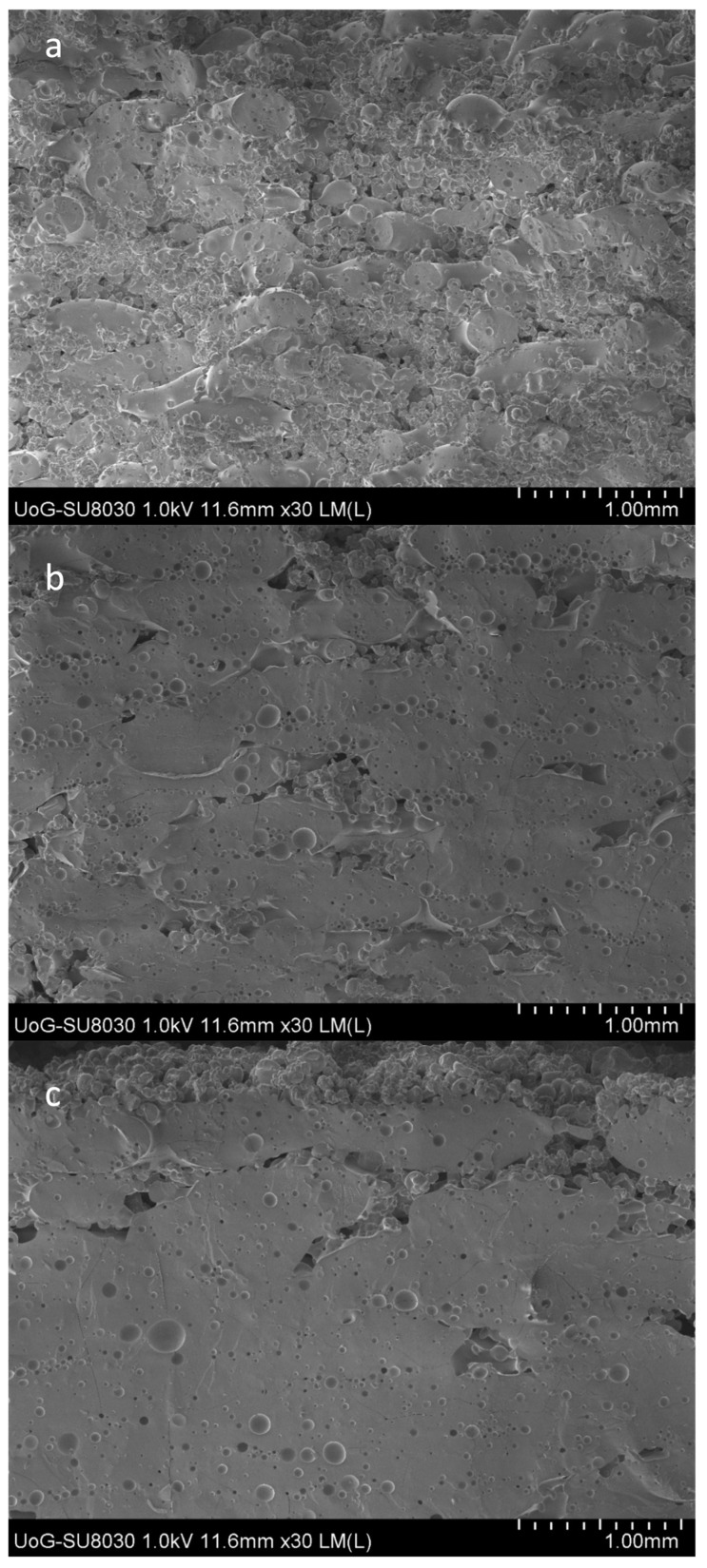
SEM images of the 3D-printed CVD/VA 64 tablets printed at (**a**) 25% (F1), (**b**) 40% (F2), and (**c**) 55% (F3) laser intensities.

**Figure 5 pharmaceutics-15-02230-f005:**
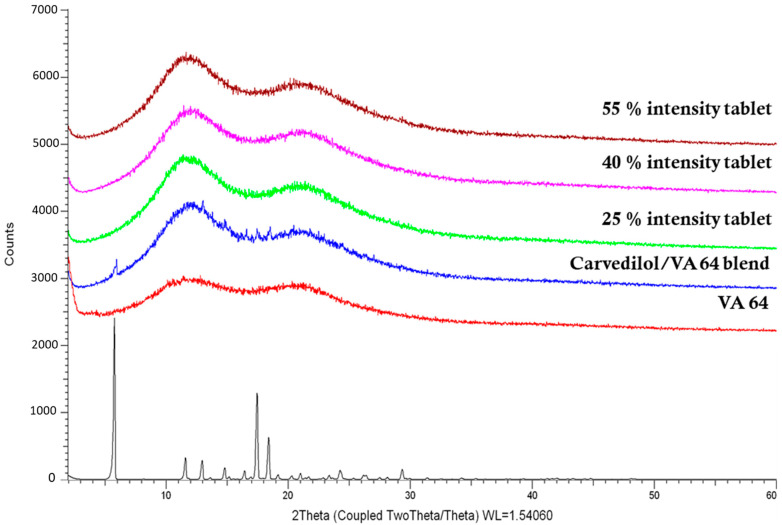
XRD graphs of the bulk materials’ physical blends for printing (3.125% CVD loading) and the respective 3D-printed tablets at different laser intensities.

**Figure 6 pharmaceutics-15-02230-f006:**
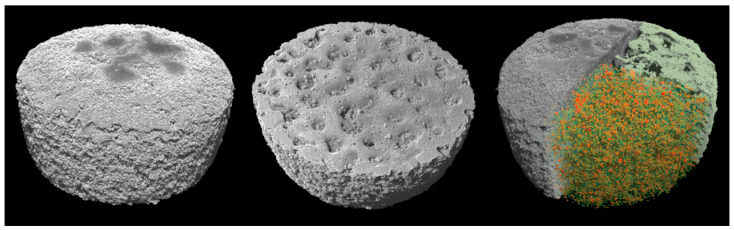
The 3D volume renderings of the SLS-printed specimens (F1) from different views. The top and bottom views are displayed on the left and middle, respectively. On the right, a clipped top view is shown which reveals both the macro-porosity and micro-porosity. Specifically, the top portion of the clipped region is selectively rendered to display the macro-porosity while the bottom portion displays the micro-porosity.

**Figure 7 pharmaceutics-15-02230-f007:**
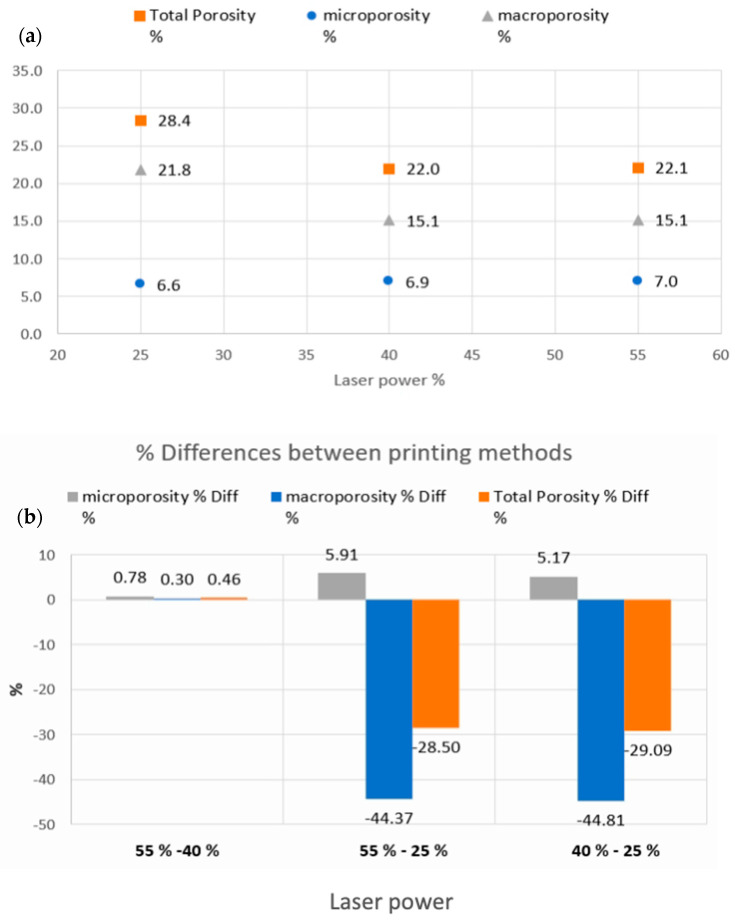
Porosity quantification for the F1–3 printed formulations. (**a**) The total porosity percentages and their constituent components of the micro- and micro-porosities that resulted from the different laser powers. (**b**) The total porosity, micro-porosity, and macro-porosity differences between the different laser power printings.

**Figure 8 pharmaceutics-15-02230-f008:**
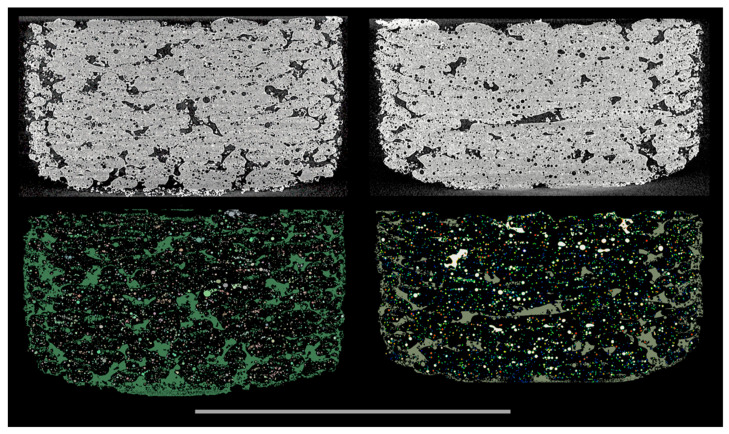
The 3D volume renderings of the SLS-printed specimens printed at 55 W laser power from different views (F3). The top and bottom views are displayed on the left and middle, respectively. On the right, a clipped top view is shown which reveals both the macro-porosity and micro-porosity. Specifically, the top portion of the clipped region is selectively rendered to display the macro-porosity while the bottom portion displays the micro-porosity. Object size: minimum Feret diameter of 5.86 mm, maximum Feret diameter of 11.47 mm, and mean Feret diameter of 9.55 mm. The scale bar is 10 mm.

**Figure 9 pharmaceutics-15-02230-f009:**
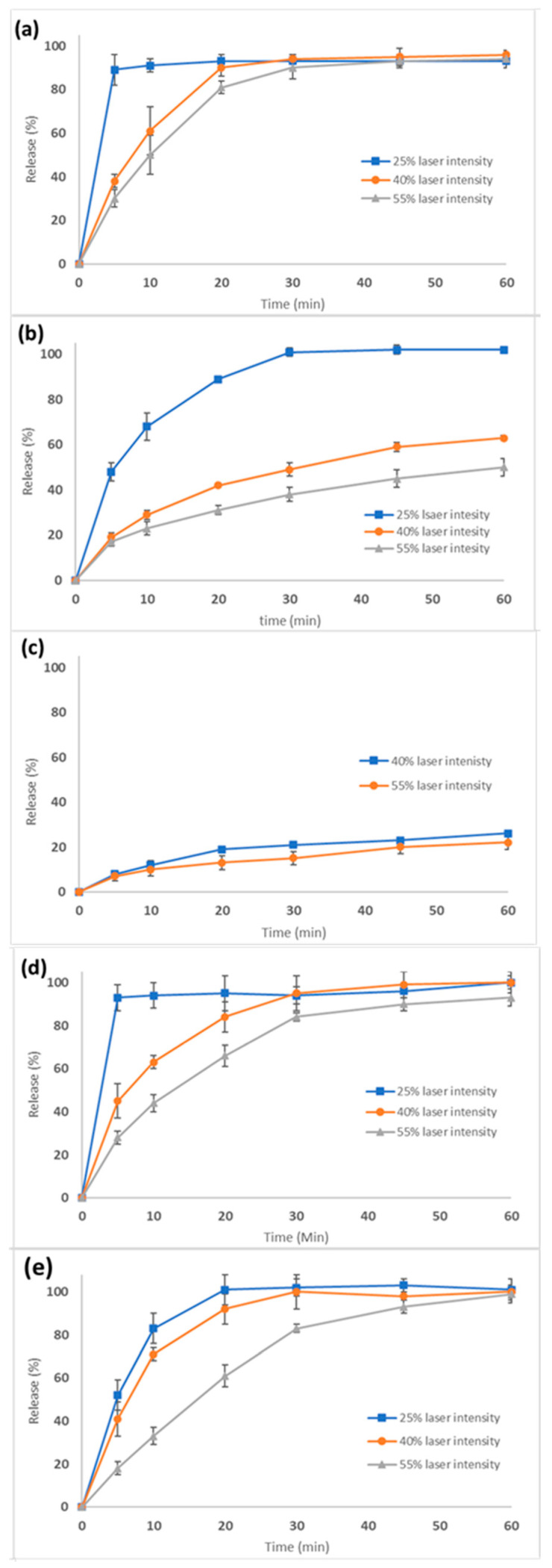
Dissolution rates of the tablets of various strengths printed at different laser intensities: (**a**) 3.125 mg, (**b**) 6.25 mg, (**c**) 12.5 mg, (**d**) 6.5 mg, and (**e**) 12.5 mg strengths.

**Table 1 pharmaceutics-15-02230-t001:** CVD content, tablet dimensions, and printing processing parameters.

No	CVD (%)	VA64 (%)	Diameter (mm)	Thickness (mm)	Intensity (%)	Tablet Weight (mg)	Tablet CVD Content (mg)
**F1**	3.125	96.875	10	4.0	25	100	3.125
**F2**	3.125	96.875	10	2.4	40	100	3.125
**F3**	3.125	96.875	10	2.2	55	100	3.125
**F4**	3.125	96.875	10	8.2	25	200	6.25
**F5**	3.125	96.875	10	4.8	40	200	6.25
**F6**	3.125	96.875	10	4.4	55	200	6.25
**F7**	3.125	96.875	10	9.4	40	400	12.5
**F8**	3.125	96.875	10	8.4	55	400	12.5
**F9**	6.25	93.75	10	4.0	25	100	6.25
**F10**	6.25	93.75	10	2.4	40	100	6.25
**F11**	6.25	93.75	10	2.2	55	100	6.25
**F12**	6.25	93.75	13	2.4	25	200	12.5
**F13**	6.25	93.75	13	1.6	40	200	12.5
**F14**	6.25	93.75	13	1.4	55	200	12.5

**Table 2 pharmaceutics-15-02230-t002:** Physical and mechanical properties of the 3D-printed CVD tablets at various CVD doses and laser intensities.

**Theoretical tablet weight** **(mg)**	100			200			400		100			200		
**Theoretical CVD** **(mg)**	3.125			6.25			12.5		6.25			12.5		
**Laser intensity** **(%)**	25	40	55	25	40	55	40	55	25	40	55	25	40	55
**Average weight** **(mg)**	98.5	101.3	103.4	210.8	209.4	205.8	403.2	405.4	102.3	103.5	101.3	207.5	204.5	208.2
**Hardness** **(N)**	10.4 ±1.4	18.9 ±1.9	24.4 ±3.7	36.5 ± 2.2	80.6 ±11.3	95.2 ± 5.4	105.4 ±8.8	125.5 ±5.9	11.4 ± 1.9	19.5 ± 2.3	26.9 ±3.6	13.5 ± 0.94	35.9 ±4.5	47.6 ±4.4
**Friability** **(%)**	10.97	1.03	0.75	14.3	1.14	0.86	1.36	0.95	11.43	0.98	0.82	12.7	1.08	0.89
**Weigh variation** **(±%)**	<1	<1	<1	<1	<1	<1	<2	<2	<1	<1	<1	<1	<1	<1

**Table 3 pharmaceutics-15-02230-t003:** Friability and dissolution rate acceptability of the 3D-printed CVD tablets at different laser intensities.

**Tablet weight (mg)**	100			200			400		100			200		
**CVD (mg)**	3.125			6.25			12.5		6.25			12.5		
**Tablet formulation**	F1	F2	F3	F4	F5	F6	F7	F8	F9	F10	F11	F12	F13	F14
**Dissolution**	**✓**	**✓**	**✓**	**✓**	**x**	**x**	**x**	**x**	**✓**	**✓**	**✓**	**✓**	**✓**	**✓**
**Friability**	**x**	**✓**	**✓**	**x**	**✓**	**✓**	**✓**	**✓**	**x**	**✓**	**✓**	**x**	**✓**	**✓**

The check marks indicate that the tablets met (a) the United States Pharmacopeia specifications in terms of dissolution and (b) the friability was <1%.

## Data Availability

The μCT data for this study is accessible and can be found on Zenodo.org with the following DOI: 10.5281/zenodo.8284414. Researchers interested in accessing the data can use this DOI to locate and download the relevant information.

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
