# Peer review of "3D Printing of Personalised Carvedilol Tablets Using Selective Laser Sintering"

_pharmaceutics, 2023, doi:10.3390/pharmaceutics15092230_

Round 1

Reviewer 1 Report

This manuscript investigated manufacturing of carvedilol personalised dosage forms of various strengths using SLS with a CO2 laser, and proved that the tuning of the SLS processing parameter laser intensity, improved printability and impacted the tablet hardness, friability, CVD dissolution rate, and the total amount of drug released. The results demonstrated that SLS is a suitable technology for the development of personalised medicines that meet the required specifications and patient 31 needs. However, there are few weaknesses that should be addressed.

1.     There are 10 figures in the manuscript, some of them can be combined into 1 figure.

2.     More details should be provided in each figure legend.

3.     Fig7 and Fig 9 need scale bars

4.     Fig 8b need standard derivation

5.     Fig 10c, no 25% laser intensity data was provided.

6.     An in-depth discussion is needed to compare your method with other currently used methods and demonstrate the advantage of your 3D printing using SLS.

Author Response

This manuscript investigated manufacturing of carvedilol personalised dosage forms of various strengths using SLS with a CO2 laser, and proved that the tuning of the SLS processing parameter laser intensity, improved printability and impacted the tablet hardness, friability, CVD dissolution rate, and the total amount of drug released. The results demonstrated that SLS is a suitable technology for the development of personalised medicines that meet the required specifications and patient 31 needs. However, there are few weaknesses that should be addressed.

  1. There are 10 figures in the manuscript, some of them can be combined into 1 figure.

We tried our best but we reduced them only to 9.

  1. More details should be provided in each figure legend.

It is done.

  1. Fig7 and Fig 9 need scale bars

Fig 7 is presented as perspective renderings, wherein objects appearing further away are intentionally rendered smaller to create a sense of depth. In such cases, traditional 2D scale bars would not be applicable, as they only work for flat, 2D renderings. To address this issue, we have provided an alternative solution in the figure legend, where we indicate the min, max and mean ferret diameter of the specimens. This should offer readers a clear understanding of the size relationships within the image.

Fig 9. The scale bars in that figure were inadvertently cropped during the grouping process. We have taken prompt action to rectify this issue by re-rendering the image with the correct scale bars. The updated Fig 9 now accurately reflects the scale of the objects depicted. Thank you for picking this up.

  1. Fig 8b need standard derivation

We want to clarify that the values shown in the figure are absolute measurements, not statistical averages. The comparisons are made between two values of two different specimens. As a result, standard deviations (stDev) are not applicable in this context.

  1. Fig 10c, no 25% laser intensity data was provided.

The reviewer is right.  This was done intentionally as the tablets were very friable to handle and not possible to print.

  1. An in-depth discussion is needed to compare your method with other currently used methods and demonstrate the advantage of your 3D printing using SLS.

Indeed, this is very good comment and have added the relative discussion after the dissolution section.

Reviewer 2 Report

The author research “3D Printing of Personalised Carvedilol Tablets Using Selective Laser Sintering” explore utilizing Selective laser sintering (SLS) for the fabrication of three-dimensional oral dosage forms for personalized medication, specially improve divided dose using Kollidon as key material. Here authors evaluated prepared dosage form for strength, friability, active dose, and dissolution rates to meet pharmacopeia specifications.

The technique is well proven and now higher focus on commercialization and application at ease. Even cost reduction is future step, besides regulatory concerns.

The observation and comments are as follows:

1.       Table 1: CVD content, tablet dimensions, and printing processing parameters, is parameters avg value? If yes, then standard deviation must be there.

2.       How much Kollidon used in batches not clear, specially, F4 to F 14? Please add column for amount of kollidon.

3.       Please cite the methods/characterization which were adopted form literature.

4.       Figure 4: Why surface is rough in few tables. Please discuss it. Is process parameter need to critically maintain?

5.       L 338: Typo error: a CO2 laser, 2 should be subscript. Look manuscript carefully for such error.

6.       Why broad range of 100-400 utilized? Will it be not possible to have 50-100 mg all tablets?

7.       Table 2: Physical and mechanical properties of 3D printed CVD tablets at various CVD 365 doses and laser intensities; same as Table 1, comment. Is n=3, if yes then SD should be there.

8.       Format error: Table 2: Physical and mechanical properties of 3D printed CVD tablets at various CVD 365 doses and laser intensities, must be bold. Similarly, 3.6…

9.       L 444: To improve the dissolution rate for the higher dosage strengths, the drug loading in the 444 CVD/VA 64 formulation was increased to 6.25%., is bit inconclusive. What author want to say.

10.   All table clarity is poor, please improve. It looks like they are pasted as image.

11.   Overall: What is novelty except technology, which is already present in market. Even material well utilized be other researchers.

12.   I would like to see, stability as per ICH guideline, as its porous structure may have issue with moisture stability along with harness data, whether it can have sustained the shocks and pressure.

Author Response

The author research “3D Printing of Personalised Carvedilol Tablets Using Selective Laser Sintering” explore utilizing Selective laser sintering (SLS) for the fabrication of three-dimensional oral dosage forms for personalized medication, specially improve divided dose using Kollidon as key material. Here authors evaluated prepared dosage form for strength, friability, active dose, and dissolution rates to meet pharmacopeia specifications.

The technique is well proven and now higher focus on commercialization and application at ease. Even cost reduction is future step, besides regulatory concerns.

The observation and comments are as follows: 

  1. Table 1: CVD content, tablet dimensions, and printing processing parameters, is parameters avg value? If yes, then standard deviation must be there.

No those are the absolute values.

  1. How much Kollidon used in batches not clear, specially, F4 to F 14? Please add column for amount of kollidon.

The reviewer is right.  We have added a column in the Table to clarify the VA64 content.  For the 6.25 and 12.5% of carvedilol the VA64 was the same. For the later we doubled the size of the tablet during printing.

  1. Please cite the methods/characterization which were adopted form literature. 

The methods adopted from literature was mainly μCT which has been well cited. In addition we have added further discussion as requested by the 3rd reviewer. All other methods are cited when similar findings are observed with our work.

  1. Figure 4: Why surface is rough in few tables. Please discuss it. Is process parameter need to critically maintain? 

This is very common feature of SLS technology as tablets are not printed with a smooth surface.

  1. L 338: Typo error: a CO2 laser, 2 should be subscript. Look manuscript carefully for such error.

It has been adjusted.

  1. Why broad range of 100-400 utilized? Will it be not possible to have 50-100 mg all tablets? 

This would be possible if the printed tablets the strength would be fixed (e.g., 3.125 or 6.25 or 12.5 only). However as stated in the discussion we aimed to print all three doses with a single blend and this is why we had to vary the size of the tablets in order to meet the drug strength.

  1. Table 2: Physical and mechanical properties of 3D printed CVD tablets at various CVD 365 doses and laser intensities; same as Table 1, comment. Is n=3, if yes then SD should be there.

The SDs have been added in Table 2.

  1. Format error: Table 2: Physical and mechanical properties of 3D printed CVD tablets at various CVD 365 doses and laser intensities, must be bold. Similarly, 3.6…

We would prefer to leave the Table as it is.  It might be confusing for the readers as all properties are interrelated with the print settings.

  1. L 444: To improve the dissolution rate for the higher dosage strengths, the drug loading in the 444 CVD/VA 64 formulation was increased to 6.25%., is bit inconclusive. What author want to say.

The reviewer is right.  We meant that by using only 3.125% in the print blend the higher CVD strength had to be printed by increasing the tablet size. However, this resulted in poor dissolution rates (Fig. 10b,c).  Hence we used a new blend (Table 1) by increasing CVD content at 6.25%.  This allowed to improve the dissolution rate for both strengths at 6.25 and 12.5 mg per tablet.

  1. All table clarity is poor, please improve. It looks like they are pasted as image. 

Thanks for picking this up. We have now added the Tables correctly.

  1. Overall: What is novelty except technology, which is already present in market. Even material well utilized be other researchers. 

      The technology has not been fully exploited for pharmaceutical dosage forms. In our case it is the first time that SLS has been developed to print personalised dosage forms at three different strengths (3.125, 6.25 and 12.5 mg) in one go and a single formulation.  In addition we clearly demonstrated that the printed tablets meet the pharmacopeia specifications which has not been demonstrated by any other group.

  1. I would like to see, stability as per ICH guideline, as its porous structure may have issue with moisture stability along with harness data, whether it can have sustained the shocks and pressure. 

      Unfortunately, this was not the scope of the work as it will require long periods for such studies. The aim of our work was to demonstrate the capabilities of SLS technology.

Reviewer 3 Report

The manuscript “3D Printing of Personalised Carvedilol Tablets Using Selective Laser Sintering” presents a very interesting study on pharmaceutical SLS 3D printing. The manuscript can be obviously published in Pharmaceutics journal. However, there are some minor issues that should be addressed.

SLS is a very promising but also very challenging technique. Probably for this reason studies on SLS printing of pharmaceuticals are still sparse. Usually, the researchers avoid showing details (especially regarding printlet structure). An advantage of the presented study is section 3.5 (uCT with quantitative analysis) as sintered tablets are intrinsically porous. Two porosity fractions called micro- and macro-porosity were assessed which is not the usual procedure. But in my opinion, there is an inconsistency between SEM and uCT. What are the SEM images in Figure 5? Using SEM you can image surfaces. The upper surface of the tablet? If yes, I think it is not representative of the entire sample. The authors stated that “The formation of non-porous printed tablets with increasing laser intensity was also observed by SEM analysis as illustrated in Fig. 5.”. But assessed porosity by uCT (Figure 8) was higher than 20%. Please clarify the issue. Moreover, it is not clear which formulations were studied by SEM and uCT.  Please use symbols FX from Table 1 to unambiguously mark studied samples in figure captions. Scale bars in Figures 7 and 9 can also be helpful.

“These results indicate that the laser power-depended porosity-change is dominated and driven by changes in the macro-porosity, which is better visualised in Fig. 9.” – is it possible to discuss the obtained porosity results in the context of the study by Kulinowski et al.? Only Kulinowski et al. (Additive Manufacturing, 2021, 38, 101761) previously characterized comprehensively pore structure in SLS printlets using uCT. They used other approach using pore thickness histograms. I’m just curious.

If it is possible, please also comment on deviations from the designed model shape as uCT reveals all defects of sintered parts (e.g. bottoms seem to be rounded).

Author Response

The manuscript “3D Printing of Personalised Carvedilol Tablets Using Selective Laser Sintering” presents a very interesting study on pharmaceutical SLS 3D printing. The manuscript can be obviously published in Pharmaceutics journal. However, there are some minor issues that should be addressed.

SLS is a very promising but also very challenging technique. Probably for this reason studies on SLS printing of pharmaceuticals are still sparse. Usually, the researchers avoid showing details (especially regarding printlet structure). An advantage of the presented study is section 3.5 (uCT with quantitative analysis) as sintered tablets are intrinsically porous. Two porosity fractions called micro- and macro-porosity were assessed which is not the usual procedure. But in my opinion, there is an inconsistency between SEM and uCT. What are the SEM images in Figure 5? Using SEM you can image surfaces. The upper surface of the tablet? If yes, I think it is not representative of the entire sample. The authors stated that “The formation of non-porous printed tablets with increasing laser intensity was also observed by SEM analysis as illustrated in Fig. 5.”. But assessed porosity by uCT (Figure 8) was higher than 20%. Please clarify the issue. Moreover, it is not clear which formulations were studied by SEM and uCT.  Please use symbols FX from Table 1 to unambiguously mark studied samples in figure captions. Scale bars in Figures 7 and 9 can also be helpful.

The reviewer is right about the differences between SEM and μCT as we had the same questions when analysed the results. The SEM image can give us a qualitative “analysis” of the tablet porosity where μCT it provides detailed results as it an advanced technology.  Fig.5 illustrates F1-3 at different laser intensities where F2-F3 look almost identical while F1 shows some porosity. TheμCT provided measurable data as the reviewer pointed out but a careful look shows that the porosity in higher laser intensities is similar. This is why we made the above claim (Fig. 7 now) that SEM and μCT results are aligned.

We have included the formulation details in SEM/μCT figures for comparison.

“These results indicate that the laser power-depended porosity-change is dominated and driven by changes in the macro-porosity, which is better visualised in Fig. 9.” – is it possible to discuss the obtained porosity results in the context of the study by Kulinowski et al.? Only Kulinowski et al. (Additive Manufacturing, 2021, 38, 101761) previously characterized comprehensively pore structure in SLS printlets using uCT. They used other approach using pore thickness histograms. I’m just curious. 

If it is possible, please also comment on deviations from the designed model shape as uCT reveals all defects of sintered parts (e.g. bottoms seem to be rounded).

At this stage, we would like to stress that in our study, we have focused on the global manifestation of micro- and macro-porosity as well as on their relative changes between the different preparation methods. For this purpose, we have separated porosity into "macro-porosity," which consists of large, interconnected voids developing across the whole volume of the specimens, and "micro-porosity," which consists of much smaller, isolated void spaces. A more in-depth characterization of the geometrical, statistical, and physical properties of these entities was outside the scope of this investigation. However, it is worth noting that an in-depth analysis of the porosity landscape, e.g. for informing Computational Fluid Dynamics (CFD) studies, is possible using various porous media analysis strategies, such as {REF1 & REF2}. In that case, pores can be handled as "particles," and specific measures such as individual size, aspect ratio, sphericity, etc., can be extracted. Equally, alternative porous media analysis strategies, such as skeletonization, can be applied to the interconnected porosity fabric (macro-porosity) if properties such as the degree of interconnectivity, tortuosity, permeability, are of interest. An example of such an approach can be seen in {REF3}, where Kulinowski et al. use such an approach to characterize the topology of the pore space.

Round 2

Reviewer 2 Report

Well, satisfied with all answers.